# Contribution of Mitochondrial DNA Heteroplasmy to the Congenital Cardiac and Palatal Phenotypic Variability in Maternally Transmitted 22q11.2 Deletion Syndrome

**DOI:** 10.3390/genes12010092

**Published:** 2021-01-13

**Authors:** Boris Rebolledo-Jaramillo, Maria Gabriela Obregon, Victoria Huckstadt, Abel Gomez, Gabriela M. Repetto

**Affiliations:** 1Center for Genetics and Genomics, Facultad de Medicina Clínica Alemana, Universidad del Desarrollo, Santiago 7710162, Chile; brebolledo@udd.cl; 2Servicio de Genética, Hospital de Pediatría Garrahan, Buenos Aires C1249ABP, Argentina; obregon.gabriela@gmail.com (M.G.O.); vickyhuckstadt@gmail.com (V.H.); abelismaelgomez@gmail.com (A.G.)

**Keywords:** 22q11.2 deletion syndrome, mtDNA heteroplasmy, congenital defects

## Abstract

Congenital heart disease (CHD) and palatal anomalies (PA), are among the most common characteristics of 22q11.2 deletion syndrome (22q11.2DS), but they show incomplete penetrance, suggesting the presence of additional factors. The 22q11.2 deleted region contains nuclear encoded mitochondrial genes, and since mitochondrial function is critical during development, we hypothesized that changes in the mitochondrial DNA (mtDNA) could be involved in the intrafamilial variability of CHD and PA in cases of maternally inherited 22q11.2DS. To investigate this, we studied the transmission of heteroplasmic mtDNA alleles in seventeen phenotypically concordant and discordant mother-offspring 22q11.2DS pairs. We sequenced their mtDNA and identified 26 heteroplasmic variants at >1% frequency, representing 18 transmissions. The median allele frequency change between a mother and her child was twice as much, with a wider distribution range, in PA discordant pairs, *p*-value = 0.039 (permutation test, 11 concordant vs. 7 discordant variants), but not in CHD discordant pairs, *p*-value = 0.441 (9 vs. 9). Only the variant m.9507T>C was considered to be pathogenic, but it was unrelated to the structural phenotypes. Our study is novel, yet our results are not consistent with mtDNA variation contributing to PA or CHD in 22q11.2DS. Larger cohorts and additional factors should be considered moving forward.

## 1. Introduction

Chromosome 22q11.2 microdeletion syndrome (22q11.2DS) is one of the most frequent rare pathogenic genomic rearrangements in humans with an estimated incidence of one in 4000 live births [1]. Clinical features include congenital heart disease in 50–70% of patients, palatal anomalies in 70–80%, developmental delay and learning difficulties in virtually all patients, increased risk of psychiatric disease, hypocalcemia, among many other characteristics [2]. Through knock-out experiments in mice, it was shown that haploinsufficiency for the transcription factor Tbx1, one of the genes located within the 22q11.2 deletion region, reproduces the cardiac outflow tract anomalies, palatal anomalies, and thymic and parathyroid hypoplasia, similar to the clinical phenotype observed in 22q11.2DS patients [3]. The syndrome is characterized by the variable expressivity of its associated phenotypes, illustrated by descriptions of phenotypic “extremes” that range from very ill newborns manifesting severe conotruncal heart disease, hypocalcemic seizures, palatal anomalies, and immunodeficiency, to individuals with a history of learning disabilities and hyper-nasal speech due to a submucous cleft palate, or adolescents or adults with schizophrenia and neither major cardiac nor palatal anomalies. Most individuals share a 22q11.2 deletion of the same size, however, deletion size does not appear to be associated with typical manifestations [2,4]. The causes of this phenotypic variability are still unknown, and may include genetic (inherited or acquired), epigenetic, environmental, or stochastic factors. Interestingly, a “two-hit hypothesis” has been proposed as a mechanism for variable expressivity for other microdeletion syndromes, where the severity of the phenotype of, for example, 15q13 and 16p12.1 microdeletions appeared to be increased in the presence of a second genomic alteration [5]. Therefore, in the search for potential second hits, some researchers have focused on characterizing the pathways in which *TBX1* was involved. Using proteomics, it was shown that the TBX1 protein upregulates expression of proteins primarily involved in metabolism and downregulates expression of proteins involved in signaling; interestingly, between 10–15% of these identified proteins localized to the mitochondria [6]. Mitochondria have their own genome, the mitochondrial DNA (mtDNA), containing only 37 genes [7]. The remaining ~1500 mitochondrial genes are nuclear encoded [8]. Therefore, the mitochondrion and nucleus must coordinate both genomes to guarantee mitochondrial function. There are six nuclear mitochondrial genes within the boundaries of the common 3Mb 22q11.2 deletion [1,9]. As described in mice, multiple energy-demanding stages of cell migration occur between E6.5 and 9.5 to form the heart [10]. In *Polg2* knockout mice (mtDNA polymerase), development stops at E8.5, around the time the linear heart undergoes looping [10,11], suggesting that proper mitochondrial function is critical for embryonic development. Similarly, it has been proposed that impaired mitochondrial function contributes to the initial establishment of the 22q11.2DS neurological phenotype early during development [12], and it was recently shown that haploinsufficiency of *MRLP40*, one of the six nuclear encoded mitochondrial genes located within the 22q11.2 deleted region, impacted mitochondrial function in iPSC-derived neurons from 22q11.2DS patients [13]. However, the role of mtDNA alleles in structural defects of 22q11.2DS has not been evaluated.

In humans, the mitochondrial DNA is inherited from the mother alone. Mitochondrial DNA mutations usually affect only a proportion of the total mitochondrial genomes, generating a state called heteroplasmy [14]. It has been observed that as the percentage of mtDNA harboring a pathogenic variant increase, mitochondrial function decline, and when energy output is insufficient, a threshold was crossed and symptoms appeared [15,16]. However, because of their non-mendelian inheritance, if mutated mtDNA genomes are present in a mother, the level of heteroplasmy inherited by her children can vary from 0% to 100% in a single generation, due to a phenomenon called the “germline bottleneck” [17,18,19], making it difficult to anticipate their effect. Recent studies have shown that the observed differences in the allele frequency for heteroplasmic variants between a mother and her child were mostly due to the germline bottleneck, and further somatic divergence only accounted for about 10% of the differences [19,20], making the measurement of non-invasive tissues, such as blood, suitable as a proxy to evaluate heteroplasmic levels during embryonic development.

The 22q11.2 deletion usually occurs as a de novo event but is found to be inherited from an affected parent in about 10% of newly diagnosed cases. Similar to unrelated cases, there is intrafamilial variation, with some members of the same family showing more severe phenotypes than others [21,22], including phenotypically discordant twins [23]. Most familial cases are inherited from affected mothers [24]. In these cases, both the 22q11.2 deleted region and the mitochondrial genome are inherited from the mother, yet phenotypic variability is often observed between a mother and her offspring [25,26]. Mitochondrial function is required for proper embryonic development, and since the 22q11.2 deletion can affect mitochondrial function, we hypothesized that additional changes in heteroplasmic mtDNA alleles observed in a child could be associated with differences in the expression of structural defects with respect to their mother. Therefore, in this study, we explored the extent of mitochondrial DNA variability among 22q11.2DS mother-child pairs discordant for congenital heart disease and palatal anomalies.

## 2. Materials and Methods

### 2.1. Subjects

Mother-child pairs with MLPA confirmed cases of 22q11.2 deletion (SALSA MLPA Probemix P250-B2 DiGeorge MRC-Holland), and available information on their cardiac and palate phenotype, were invited to participate.

Mothers and their children were initially evaluated by a clinical geneticist at Hospital Garrahan. Children suspected of having cardiac or palate anomalies were further evaluated by a cardiologist or an ENT doctor, accordingly. The cardiac and palate phenotype of mothers was obtained from their medical history. Regarding the heart, all structural abnormalities were considered. Regarding the palate, a patient was considered normal if none of the following were present: submucous or posterior cleft palate, bifid uvula, hyper nasal voice, or gastroesophageal reflux.

Sample phenotype description is shown in Table 1. Pairs were considered to be phenotypically “concordant”, when both members shared the presence or absence of any cardiac or palatal structural anomalies, and “discordant”, when one of the two had anomalies and the other member did not.

### 2.2. DNA Extraction and Sequencing

Total DNA from blood was submitted for 2 × 100 DNBSeq™ Small genome sequencing at BGI Americas Corporation. According to BGI’s protocol, mitochondrial DNA was enriched using a long-range PCR reaction with primers: forward (MT:15149-15174) TGAGGCCAAATATCATTCTGAGGGGC, and reverse (MT:14816-14841) TTTCATCATGCGGAGATGTTGGATGG, Phusion High Fidelity DNA Polymerase and 5x Phusion GC buffer (New England Biolabs). 

### 2.3. Mitochondrial DNA (mtDNA) Variant Calling

The heteroplasmy variant calling pipeline has been previously published [18]. In short, raw sequencing reads were aligned with bwa [27] against the hg19 version of the human genome but replacing the mitochondrial genome with the revised Cambridge Reference Sequence (rCRS, NC_012920). Alignments were refined using Samtools [28] for manipulation, Picard Tools for deduplication, GATK for base quality score recalibration [29], and bamleftalign, from the Freebayes package, for indel standardization [30]. Improved bam formatted files were processed with a custom script that utilized the Naïve Variant Caller and Allele Counts tools [31] to extract per nucleotide allele counts for all 16,569 human mtDNA positions. High-confidence heteroplasmic sites were defined as those sites with the following: (1) depth ≥ 1000, (2) minor allele frequency ≥ 1%, (3) no strand bias, (4) outside known problematic regions (303–311, 3107, 16,185–16,193), (5) no position-in-read bias, and (6) statistically significant in a Poisson test comparing allele frequency to error rate at the position among all the other samples.

### 2.4. mtDNA Haplogroup Assignment

We extracted the most common allele for each position in the mtDNA of a sample based on the count of A’s, C’s, G’s, and T’s at a position. Then, we concatenated all alleles into a single string of 16,569 letters to create a major-allele sequence for each sample in fasta format. Then, haplogroup assignment was calculated on a sample’s major allele fasta file using the web version of Haplogrep at https://haplogrep.uibk.ac.at [32].

### 2.5. Comparison of Concordant and Discordant Mother-Child Pairs

We counted the number of high-confidence heteroplasmic sites observed in a mother and her child, calculated the difference between them as ∆NS = number of sites in child − number of sites in mother, and compared concordant and discordant pairs. 

We tabulated each heteroplasmic site in pairs and considered a site to be transmitted if the mother had at least 10 reads supporting the high-confidence heteroplasmic allele observed in her child (Appendix A). Then, we calculated the allele frequency change (∆AF) between a mother and her child as ∆AF = AF_child_ − AF_mother_ and compared the distribution of ∆AF values among concordant and discordant pairs. 

### 2.6. Statistical Analyses and Results Reproducibility

Descriptive statistics are presented and indicated as mean ± s.d. or median and range. The Shapiro–Wilk test was used to assess normality. The *p*-value for the comparison of concordant and discordant pairs was calculated with a 1000 replicas permutation test of the Mann–Whitney U statistic. Results were considered to be statistically significant if *p* ≤ 0.05. A Jupyter notebook [33,34] with the code demonstrating all the steps in data processing, statistical analysis, and figure generation is available at https://github.com/berebolledo/mtDNA22q.

## 3. Results

### 3.1. Subjects

Thirty-two patients participated in the study, 15 mothers and 17 children, corresponding to 17 mother-offspring pairs with the 22q11.2 microdeletion. Two mothers (DG184 and DG220) had two children each with the 22q11.2 deletion. The median age of the mothers was 29 years (range 19–42), and six months (range one month to nine years) for the children. Among the children, there were nine girls (53%), and eight boys (47%). Two mothers (13.3%) had a CHD, while this feature was present in 13/17 (76.5%) of the children. Six mothers (40%), and 13/17 (76.5%) children had a diagnosis of palate anomalies. Mother DG237 and her child DG236 were excluded from the remaining analyses because they did not share the same deletion size, therefore, it could not be assumed that the 22q11.2 deletion was inherited, despite their maternal relationship confirmed by their mtDNA haplogroup (Appendix A). Finally, for the cardiac phenotype, there were five concordant mother-child pairs, and 11 discordant pairs. According to the palatal phenotype, there were seven concordant mother-child pairs, and nine discordant pairs. A summary of palatal and cardiac manifestations is presented in Table 1.

### 3.2. Sequencing Results

On average, 16,326 ± 90 mtDNA positions were captured per sample (mean ± s.d.), with a median depth of 47,000× (Appendix A). We identified a particular drop in sequencing depth across all samples around position MT:15000 consistent with the location of the primers used for mtDNA enrichment (Appendix A, Methods). Therefore, we excluded the region MT:14800-15200 from the remaining analyses.

### 3.3. mtDNA Variants in Mother/Child Pairs

Mitochondrial DNA haplogroups were determined with Haplogrep [32]. The average probability of haplogroup assignment was 0.9 ± 0.07 (mean ± s.d.). Eight mother-offspring pairs had haplogroups mostly observed in Native American populations (haplogroups A, B, and D), and eight pairs had haplogroups of European origin (haplogroups H, T, and U) (Appendix A). We confirmed all mother-child pair maternal relationships based on their haplogroup, and their mtDNA sequence pairwise distance in a neighbor joining tree (Appendix A).

Heteroplasmic mtDNA variants were calculated as described in [18]. We found 26 high-confidence sites with minor allele frequency (MAF, i.e., the frequency of the second most common allele observed in an individual) ≥1% among 30 samples (median number of sites per sample 1, range 0–5), representing 18 transmission events (see Methods). The unusual MAF distribution of samples DG224 and DG33 (Appendix A), both with five sites each, raised the suspicion of possible sample contamination. Consequently, we evaluated this possibility using a phylogenetic approach [35] and found no evidence of sample cross-contamination for either sample (Appendix A).

These 26 sites were distributed as follows: five in the D-loop, 19 in nine protein coding genes, and two in RNA genes (Table 2). Only one of the variants was considered to be “confirmed pathogenic” according to MITOMAP [36] and the Mitochondrial Disease Sequence Data Resource Consortium (MSeqDR) [37], change m.12315G>A in the *MT-TL2* gene, found in sample DG224 was associated with Kearns–Sayre syndrome and risk of atherosclerosis [38].

### 3.4. Comparison of Concordant and Discordant Pairs

Since children inherit both the 22q11.2 deletion and the mitochondrial genome from their mother, we hypothesized that changes in the number or allele frequency of mtDNA variants could be associated with their phenotype discordance. First, we calculated the difference in the number of high-confidence heteroplasmic sites between a mother and her child (∆NS, see Methods), and observed that cardiac concordant pairs had ∆NS = 1 (range 0–4) and cardiac discordant pairs had ∆NS = 0 (range −1–2). Similarly, palate concordant pairs had ∆NS = 0 (range −1–4) and palate discordant pairs had ∆NS = 0 (range −1–1). The comparison of ∆NS for the cardiac phenotype was statistically significant *p* = 0.012, but not for the palatal phenotype, *p* = 0.073.

Next, we calculated the allele frequency change (∆AF) between a mother and her child for the 18 heteroplasmic alleles transmissions identified (Methods and Appendix A). First, we compared ∆AF of all transmissions, regardless of the phenotype status, against ∆AF from a published cohort of 39 healthy mother-child pairs of European decent contributing 98 transmissions [18]. The ∆AF followed an approximately normal distribution centered on zero (Figure 1A) median ∆AF healthy = −0.004 (range −0.440–0.409) vs. median ∆AF 22q11.2DS = −0.004 (range −0.588–0.241), *p*-value = 0.687), reflective of genetic drift as the main contributor to the allele frequency variability [18,19].

Then, we compared concordant and discordant 22q11.2DS pairs. For the cardiac phenotype (Figure 1B), heteroplasmic variants from concordant pairs (*n* = 9) had median ∆AF = 0.01 (range −0.60–0.09), and heteroplasmic variants from discordant pairs (*n* = 9) had median ∆AF = −0.01 (range −0.20–0.24), *p*-value = 0.441. For the palate phenotype (Figure 1C), heteroplasmic variants from concordant pairs (*n* = 11) had median ∆AF = 0.01 (range −0.03–0.02), and heteroplasmic variants from discordant pairs (*n* = 7) had median ∆AF = −0.02 (range −0.60–0.1), *p*-value = 0.039.

## 4. Discussion and Conclusions

This study analyzed the contribution of mitochondrial DNA variability to the intrafamilial variation in cardiac and palatal anomalies observed in 22q11.2 deletion syndrome patients. Evaluating the contribution of mitochondrial DNA variants to the cardiac or palatal phenotypes is novel, since mitochondrial dysfunction has been primarily studied in the context of the neurological phenotypes associated with the 22q11.2DS [9,13,39], disregarding the mitochondria’s pivotal role during the development of multiple other structures [10,11].

The frequency of congenital cardiac and palatal anomalies in our patients reflected the frequency observed in similarly ascertained 22q11.2DS cohorts [1]. As expected, mothers had lower frequencies of anomalies as compared with their offspring, likely due to ascertainment effects, and the lower survival and reproductive fitness observed in individuals with severe heart defects [40]. Our bioinformatic pipeline took into account multiple sources of error (i.e., batch sequencing errors, strand bias, cycle bias, nuclear pseudogenes, or NUMTs). This pipeline has been applied before, and the validity of mtDNA sequencing results has been well demonstrated by us and others using Sanger sequencing and droplet digital PCR [18,41]. Therefore, despite not performing additional validation for this study, we believe our results reflect true positive variants. This was also confirmed by the presence of the allele in both the mother and her child. We observed a significant difference in the number of high-confidence heteroplasmic sites between concordant and discordant pairs affected by congenital heart disease, and the distribution of ∆AF between concordant and discordant pairs affected by palatal anomalies. These results are consistent with the hypothesis that mtDNA heteroplasmy could contribute to the incomplete penetrance of the palatal phenotype. However, the differences were driven by two extreme cases where affected children had much lower allele frequency than their unaffected mothers, i.e., pair 5 DG132/DG133 m.9507T>C and pair 11 DG220/DG222 m.16290C>T. However, the allele frequency of DG221, another child of DG220, who was also affected, had the opposite direction, thus, rendering this allele unrelated to the palatal phenotype. We observed more nonsynonymous than synonymous variants (12 vs. 7), but none of them were described as pathogenic in MITOMAP [36]. It has been recently reported that a particular haplotype of non-pathogenic mitochondrial missense variants segregated in two families with Leber’s hereditary optic neuropathy (LHON), suggesting that not necessarily a single pathogenic variant, but multiple missense variants could lead to reduced mitochondrial function and a clinical phenotype, therefore, we believe that the contribution of non-pathogenic missense mtDNA variants should be further explored [42]. Only the variant m.12315G>A, located in the *MT-TRNL2* gene, was described as pathogenic. This variant was present at a low level in unaffected concordant pair 12, thus, it was unrelated to the cardiac or palatal phenotypes.

Our results could be limited by the small sample size, consistent with 22q11.2DS as a rare disorder with an inheritance rate of 10% and possible reproductive constraints [24]. Nevertheless, to our knowledge, our study constitutes the largest ever analysis of mtDNA heteroplasmy in maternally transmitted 22q11.2DS. We expect our results to motivate the analysis of mtDNA in larger cohorts of maternally transmitted 22q11.2DS, and to also consider the contribution of the paternal 22q11.2 genes to the phenotypic differences. In fact, another source of mitochondrial variation that would be worth exploring in the context of complex phenotypes is the so-called “mito-nuclear discordance” (MND). MND refers to the difference in the ancestral origin of the mtDNA and nuclear mitochondrial genes, and it is known to cause changes in OXPHOS efficiency in model organisms [43]. Whether these incompatibilities naturally exist, and the paternal 22q11.2 allele could contribute to it, is currently unknown, but some signals of positive selection for ancestry concordance of nuclear and mtDNA alleles have been found in African American and Puerto Rican populations, and should be considered moving forward [44].

## Figures and Tables

**Figure 1 genes-12-00092-f001:**
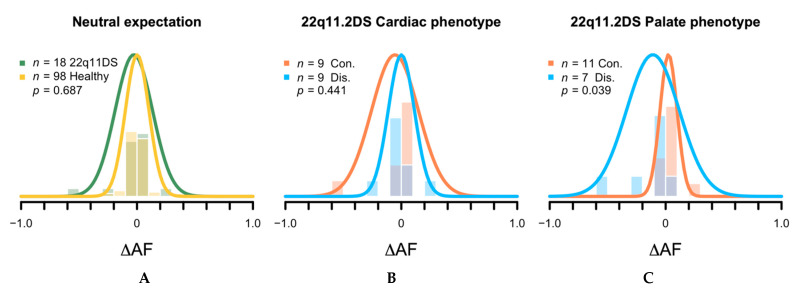
Heteroplasmic allele frequency change (∆AF) between a mother and her child. The histograms describe the actual distribution of the data, whereas the curves represent the expected density distribution provided each sample’s mean and standard deviation. (**A**) Comparison of the distribution of ∆AF between healthy individuals and 22q11.2DS patients. Comparison of the distribution of ∆AF between phenotypically concordant and discordant mother-child pairs for the cardiac (**B**) and palatal (**C**) phenotypes. Con, concordant pairs and Dis, discordant pairs.

**Table 1 genes-12-00092-t001:** Clinical phenotype of patients included in this study.

Pair	Sample	Relationship	Deletion Size (Mb)	Palate	Cardiac
1	DG18	child	3.0	VPI/BU	TA
	DG213	mother	3.0	normal	MVP
2	DG33	child	3.0	VPI	normal
	DG34	mother	3.0	VPI	normal
3	DG250	child	3.0	VPI	IIA
	DG60	mother	3.0	VPI/BU	normal
4	DG233	child	3.0	VPI	ASD
	DG76	mother	3.0	normal	normal
5	DG132	child	3.0	VPI	VSD
	DG133	mother	3.0	normal	normal
6	DG139	child	1.5	VPI	TOF
	DG161	mother	1.5	normal	normal
7	DG179	child	3.0	normal	TA
	DG180	mother	3.0	normal	normal
8	DG185	child	3.0	VPI	VSD
	DG184	mother	3.0	normal	normal
9	DG193	child	3.0	VPI	ASD
	DG184	mother	3.0	normal	normal
10	DG221	child	3.0	VPI	normal
	DG220	mother	3.0	normal	normal
11	DG222	child	3.0	VPI	normal
	DG220	mother	3.0	normal	normal
12	DG224	child	3.0	normal	normal
	DG225	mother	3.0	normal	normal
13	DG231	child	3.0	SMCP	TOF
	DG232	mother	3.0	VPI	Murmur
14	DG236 *	child	1.5	normal	TA
	DG237 *	mother	3.0	normal	VSD
15	DG242	child	3.0	VPI	TA
	DG243	mother	3.0	VPI	normal
16	DG246	child	3.0	VPI	VSD
	DG247	mother	3.0	CP	normal
17	DG249	child	3.0	normal	TA
	DG248	mother	3.0	CP	normal

ASD, atrial septal defect; BU, bifid uvula; CP, cleft palate; IIA, interrupted aortic arch; MVP, mitral valve prolapse; SMCP, submucous cleft palate; TOF, tetralogy of Fallot; TA, truncus arteriosus; VPI, velopharyngeal insufficiency; VSD, ventricular septal defect. * Excluded due to discordance in deletion size (see text).

**Table 2 genes-12-00092-t002:** High quality heteroplasmic sites.

Pair	Sample	Class	HGVS	Mj;Mn	M.A.F	Depth	*p*-Value	Gene	Effect	AF Mitomap
2	DG33	c	m.10873T>C	C;T	0.014	11859	5.0 × 10^−147^	*ND4*	p.Pro38=	0.336
	DG33	c	m.14668C>T	T;C	0.011	4713	2.0 × 10^−82^	*ND6*	p.Met2=	0.041
	DG33	c	m.14783T>C	C;T	0.011	13425	3.0 × 10^−67^	*CYTB*	p.Leu13=	0.212
	DG33	c	m.15301G>A	A;G	0.019	10341	2.0 × 10^−23^	*CYTB*	p.Leu185=	0.287
	DG33	c	m.15326A>G	A;G	0.022	14117	2.0 × 10^−13^	*CYTB*	p.Thr194Ala	0.987
	DG34	m	m.6802A>G	A;G	0.014	37167	8.0 × 10^−302^	*COX1*	p.Asp300Gly	-
4	DG76	m	m.15786T>C	T;C	0.013	66261	6.0 × 10^−04^	*CYTB*	p.Phe347Ser	-
5	DG132	c	m.9507T>C	T;C	0.152	55188	3.0 × 10^−62^	*COX3*	p.Phe101Leu	-
	DG133	m	m.9507T>C	T;C	0.356	61348	0.0	*COX3*	p.Phe101Leu	-
6	DG161	m	m.5492T>C	T;C	0.012	29462	0.0	*ND2*	p.Pro341=	0.003
7	DG179	c	m.150C>T	C;T	0.256	19122	4.0 × 10^−123^	Dloop	-	0.134
	DG179	c	m.1316T>C	T;C	0.105	21577	6.0 × 10^−09^	*RNR1*	-	-
	DG179	c	m.5054G>A	G;A	0.02	70310	0.0	*ND2*	p.Pro195=	0.004
	DG180	m	m.150C>T	C;T	0.015	19275	1.0 × 10^−92^	Dloop	-	0.134
8	DG184	m	m.15591G>A	G;A	0.023	26908	0.0	*CYTB*	p.Arg282Gln	-
11	DG222	c	m.16290C>T	T;C	0.32	13735	2.0 × 10^−66^	*Dloop*	-	0.039
12	DG224	c	m.6190G>A	G;A	0.011	8443	2.0 × 10^−126^	*COX1*	p.Arg96His	-
	DG224	c	m.10075T>C	T;C	0.021	77194	6.0 × 10^−120^	*ND3*	p.Ile6Thr	-
	DG224	c	m.12315G>A	G;A	0.011	44701	2.0 × 10^−281^	*TRNL2*	Pathogenic	-
	DG224	c	m.12457G>A	G;A	0.012	2796	1.0 × 10^−79^	*ND5*	p.Ala41Thr	-
	DG224	c	m.16182A>C	C;A	0.155	2721	4.0 × 10^−133^	Dloop	-	0.065
	DG225	m	m.16182A>C	C;A	0.188	2299	4.0 × 10^−86^	Dloop	-	0.065
13	DG231	c	m.4136A>G	A;G	0.191	34006	1.0 × 10^−03^	*ND1*	p.Tyr277Cys	0.001
	DG232	m	m.4136A>G	A;G	0.159	38779	3.0 × 10^−04^	*ND1*	p.Tyr277Cys	0.001
15	DG243	m	m.15431G>A	G;A	0.018	38994	2.0 × 10^−23^	*CYTB*	p.Ala229Thr	0.018
17	DG249	c	m.9941A>G	A;G	0.02	79897	2.0 × 10^−104^	*COX3*	p.Val245=	0.001

Class c, child; Class m, mother; Mj, major allele; Mn, minor allele; M.A.F, minor allele frequency; *p*-value, Poisson test comparing error rate and frequency. Patho, confirmed pathogenic variant; AF Mitomap, allele frequency in Mitomap.

## Data Availability

Raw sequencing data is available under Bio Project accession number PRJNA636010.

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
