# Peer review of "Contribution of Mitochondrial DNA Heteroplasmy to the Congenital Cardiac and Palatal Phenotypic Variability in Maternally Transmitted 22q11.2 Deletion Syndrome"

_genes, 2021, doi:10.3390/genes12010092_

Round 1
Reviewer 1 Report
The authors present an investigation into the possible involvement of mtDNA variants in the expression of cardiac dysfunction or palatal defects in 22q11.2 deletion syndrome.
The methodology is sufficient, though blood is often not a reliable tissue for mtDNA analyses and this should be remarked upon in the text. Moreover, the variants that are reported do not demonstrate marked skewing between mother and child - are exclusively present at a low level of heteroplasmy. To consider the involvement of a mtDNA variant, levels would have to be significantly higher - in children, disease-associated mtDNA variants are typically present at high levels even in blood.
I think the idea was a good one, but I don't agree with the data support a link between mtDNA variants and penetrance in these cases; there is no evidence of increased frequency of palatal defects in mitochondrial disease patients in the literature or in the clinic. Moreover, there is no enrichment for pathogenic mitochondrial disease variants within the sequences reported here. Given that the vast majority of variants are synonymous so I am unsure what pathomechanism was being proposed?
Indeed 133/132 where there is a shift in hereroplasmy, there is a reduction in the affected child compared to the mother who has a higher level of COX variant.
I would take the risk of atherosclerosis and the m.12315G>A variant with a pinch of salt to be honest!
Whilst not strictly a mitochondril protein, other groups have reported second hits in the TANGO2 gene in numerous patients with cardiac involvement; a literature search has not demonstrated an enrichment of structural cardiac defects, but it does validate that additional phenotypes are likely to have a second hit involved, present either on the non-tranmitted maternal allele (in the case of an affected mother, and asymptomatic child) or on the paternally-inherited allele in children who are affected with asymptomatic mothers (with respect to the phenotype of interest).
I would suspect that the palatal or cardiac involvement is due to an underlying defect in another gene and not mtDNA variants and that this should be considered a negative result - that said, I believe it is still worthy of publication because the idea was a good one. Screening for a second hit in the deleted region could be a worthy venture, admittedly beyond the scope of this study.
Minor comment - please classify the missense variants, proving the protein consequence and minimally the population frequency from Mitomap.
Rename variants according to HGVS guidelines (e.g m.12315G>A) and rename the pathogenic variant gene to MTTL2 (or MT-TL2 if you prefer)
Do you consider the variants to be genuinely present rather than clonal amplifaction of a PCR artefact? Did you repeat the PCR and library prep/sequencing, or were any validated using an alternative method e.g. fluorescent PCR-RFLP?
Reviewer 2 Report
The article examines the role of mtDNA variants transmission rate changes as a possible modifiers of 22q11 phentype. The analytical framework based on maternal inheritance
and bottleneck effect is interesting, novel and worth further exploration. The weak point of the presented research is the low number of mother-offspring
pairs due to rarity of the disease. The authors are aware of it and raise that issue in the discussion. Statistical significant result in patients with
palatal abnormality has to be treated with caution and should be replicated in future.
Below please find my remarks:
1. The clinical criteria palatal anomalies are not provided. How was defined velopharyngeal insufficiency and how it was excluded in patients described as normal. Lack of palatal abnormaliteies in the mothers contributed to the statistically significant result for the palate. Mitral valve prolapse is not an anomaly typical 22q11 and might be a phenocopy. Therefore the number of discorcondant pairs with CHD might have been insuffcient to detect association
2. The MLPA kit used is not provided. Does it discriminate between all the deletion lengths?
3. MAF should be explained as it might be confused with population frequency at first look. What is the rational for threshold selection (i.e. MAF 1%).
How confident we can be that changes in MAF detected by sequencing reflect the biological truth?
4. Could the author formulate a biological hypothesis explaining the results. Changes in transmission could modify the phenotype in both directions (protective vs damaging).
The pair 17 is an example of a reduction in severity from mother (CP) to child (normal). Is this pair different from the others in means of transmission
rate?
5. Color coding scheme of the histogram on fig 1 is not clear to me.
6. In my opinion the discussion should mention a need for replication of the results in a larger cohort of 22q patients
Author Response
Dear reviewer,
Please, see the attachment.
